# Potent Activation of Human but Not Mouse TRPA1 by JT010

**DOI:** 10.3390/ijms232214297

**Published:** 2022-11-18

**Authors:** Masaki Matsubara, Yukiko Muraki, Noriyuki Hatano, Hiroka Suzuki, Katsuhiko Muraki

**Affiliations:** Laboratory of Cellular Pharmacology, School of Pharmacy, Aichi-Gakuin University, 1-100 Kusumoto, Chikusa, Nagoya 464-8650, Japan

**Keywords:** JT010, calcium channel, transient receptor potential ankyrin repeat 1, synoviocytes, dorsal root ganglion

## Abstract

Transient receptor potential (TRP) ankyrin repeat 1 (TRPA1), which is involved in inflammatory pain sensation, is activated by endogenous factors, such as intracellular Zn^2+^ and hydrogen peroxide, and by irritant chemical compounds. The synthetic compound JT010 potently and selectively activates human TRPA1 (hTRPA1) among the TRPs. Therefore, JT010 is a useful tool for analyzing TRPA1 functions in biological systems. Here, we show that JT010 is a potent activator of hTRPA1, but not mouse TRPA1 (mTRPA1) in human embryonic kidney (HEK) cells expressing hTRPA1 and mTRPA1. Application of 0.3–100 nM of JT010 to HEK cells with hTRPA1 induced large Ca^2+^ responses. However, in HEK cells with mTRPA1, the response was small. In contrast, both TRPA1s were effectively activated by allyl isothiocyanate (AITC) at 10–100 μM. Similar selective activation of hTRPA1 by JT010 was observed in electrophysiological experiments. Additionally, JT010 activated TRPA1 in human fibroblast-like synoviocytes with inflammation, but not TRPA1 in mouse dorsal root ganglion cells. As cysteine at 621 (C621) of hTRPA1, a critical cysteine for interaction with JT010, is conserved in mTRPA1, we applied JT010 to HEK cells with mutations in mTRPA1, where the different residue of mTRPA1 with tyrosine at 60 (Y60), with histidine at 1023 (H1023), and with asparagine at 1027 (N1027) were substituted with cysteine in hTRPA1. However, these mutants showed low sensitivity to JT010. In contrast, the mutation of hTRPA1 at position 669 from phenylalanine to methionine (F669M), comprising methionine at 670 in mTRPA1 (M670), significantly reduced the response to JT010. Moreover, the double mutant at S669 and M670 of mTRPA1 to S669E and M670F, respectively, induced slight but substantial sensitivity to 30 and 100 nM JT010. Taken together, our findings demonstrate that JT010 potently and selectively activates hTRPA1 but not mTRPA1.

## 1. Introduction

Transient receptor potential (TRP) ankyrin repeat 1 (TRPA1), which is activated by noxious cold, mechanical stress, irritant chemicals, and/or clinical drugs, is widely expressed in neuronal and non-neuronal organs [1,2,3,4,5,6,7,8]. Transgenic mice lacking TRPA1 are less sensitive to mechanical stimulation, cold stimuli, and tumor necrosis factor α-induced mechanical hyperalgesia, suggesting that TRPA1 is a nociceptor that mediates acute and inflammatory pain [9,10,11,12,13,14]. Our recent study revealed that 9,10-phenylquinone (9,10-PQ), the most potent chemical that activates nuclear factor erythroid 2-related factor 2 among 1395 mainstream cigarette smoke components, effectively activates human TRPA1 [15,16], indicating that chemical compounds, including 9,10-PQ, activate TRPA1 as environmental irritants [17].

The chemical compound JT010, 2-chloro-N-(4-(4-methoxyphenyl)thiazol-2-yl)-N-(3-methoxypropyl-acetamide, developed by Takaya et al., potently and selectively activates TRPA1 [18]. The half maximum concentration required for TRPA1 activation was reported to be 0.65 nM when applied to human embryonic kidney (HEK) cells expressing human TRPA1 (hTRPA1). In contrast, JT010, even at 1 μM, did not activate TRP vanilloid family type 1, 3, and 4 (TRPV1, TRPV3, TRPV4), TRP melastatin family type 2 and 8 (TRPM2, TRPM8), and TRP canonical family type 5 (TRPC5), suggesting that JT010 is a useful TRPA1 activator as a potential pharmacological tool. Several TRPA1 cysteine residues are targeted by physiological and non-physiological electrophilic compounds that activate the channel [2,19]. Particularly, recent studies have revealed that as a two-step model, cysteine 621 (C621) is critical for channel activation by electrophilic compounds and cysteine 665 (C665) is supportive and important for activation [20,21]. Indeed, hTRPA1 activation by 9,10-PQ is dependent on both C621 and C665 at the N-terminus of the channel [16]. It has also been shown that the C621 mutation in hTRPA1 prevents channel activation by JT010 [18]. Moreover, phenylalanine 669 (F669) is critical for the binding of JT010 to hTRPA1 [22]. Meanwhile, it has been proposed that the attachment of a large electrophile, such as JT010, to C621 is sufficient for the full activation of the channel. Moreover, confirmation of the binding pocket is supported by the interaction between lysine 671 (K671) and the C terminus of the TRP helix of TRPA1 [21]. In contrast, non-electrophilic compounds such as Δ9-tetrahydrocannabinol, nicotine, and menthol activate TRPA1 via different mechanisms [1,23,24]. Therefore, pharmacological tools are important to understand the physiological functions of TRPA1, and extensive efforts have been made to develop highly selective TRPA1 agonists and antagonists.

In this study, we showed that a potent TRPA1 agonist, JT010, at concentrations under 100 nM can activate hTRPA1 but not mTRPA1. A comparison of the JT010-induced response of wild-type and mutant TRPA1s in humans and mice in response to allyl isothiocyanate (AITC) revealed that the mutation of hTRPA1 at position 669 from phenylalanine to methionine (F669M) reduced the sensitivity to JT010, and the double mutation at S669 (serine) and M670 (methionine) of mTRPA1 to glutamine (S669E) and phenylalanine (M670F), respectively, induced weak but substantial sensitivity to JT010. Treatment with JT010 also effectively activated endogenous hTRPA1 in human fibroblast-like synoviocytes (FLSs) with inflammation, but not mTRPA1 in dorsal root ganglion (DRG) cells isolated from mice. Our findings provide novel and important pharmacological evidence that JT010 is a much weaker TRPA1 agonist in mice than in humans.

## 2. Results

To confirm the expression of wild-type and mutant TRPA1s in human embryonic kidney (HEK) cells, the TRPA1 channel function was tested by applying AITC at the end of each experiment, except during the application of high JT010 concentrations [25]. Additionally, we confirmed the expression of wild-type and mutant hTRPA1 by Western blotting at the protein level (Appendix A) but failed to find any specific antibody against mTRPA1 (three different antibodies used, not shown). Using HEK cells expressing wild-type hTRPA1 (HEK-hTRPA1), we examined the effects of JT010 on hTRPA1. As shown in Figure 1A, treating HEK-hTRPA1 cells with JT010 at concentrations ranging from 0.3 to 100 nM effectively induced a Ca^2+^ response, whereas the treatment failed to evoke any Ca^2+^ response in control HEK cells (Appendix A). The dose–response relationship observed indicated that the half maximum concentration (EC_50_) required for 50% response was approximately 10 nM (Figure 1B), suggesting that JT010 is a potent TRPA1 agonist [18]. To confirm whether JT010 also induces the Ca^2+^ response of mTRPA1, we applied JT010 to HEK cells expressing wild-type mTRPA1 (HEK-mTRPA1). Surprisingly, applying 10 nM JT010 induced a small Ca^2+^ response of mTRPA1 compared with that in control HEK cells (Figure 1D vs. Appendix A), whereas 100 μM AITC elicited a large response of mTRPA1 and hTRPA1 (Figure 1C,D). Application of much higher JT010 (1000 nM) concentrations induced a moderate Ca^2+^ response of mTRPA1 (Appendix A), suggesting that mTRPA1 is much less sensitive to JT010 than hTRPA1.

To further examine whether JT010 potently activates hTRPA1 but not mTRPA1, we applied 10–100 nM JT010 to HEK-hTRPA1 and HEK-mTRPA1 cells in whole-cell recording mode (Figure 2). To maintain TRPA1 channel activity during recording, we applied chemical agents in the absence of external Ca^2+^ in the standard HEPES-buffered bathing solution (SBS) and the presence of internal Ca^2+^ at 0.3 μM in a pipette solution [16]. As shown in Figure 2A–C, exposure of HEK-hTRPA1 cells to JT010 induced inward and outward currents at −90 and +90 mV, respectively, in a concentration-dependent manner. Moreover, a TRPA1 antagonist, A-967079 (A96), at 5 μM abolished JT010-induced currents, demonstrating that JT010 potently activated hTRPA1 channel currents. In contrast, the application of JT010 to HEK-mTRPA1 cells did not induce any clear mTRPA1 channel currents, while 30 μM AITC markedly induced currents at −90 and +90 mV, sensitive to A96 (Figure 2D–F). Moreover, to exclude the possibility that relatively higher intracellular Ca^2+^ (0.3 μM) modifies the effects of JT010 on mTRPA1, HEK cells were internally superfused with only 1 mM EGTA without Ca^2+^ (Appendix A). Even under these experimental conditions, JT010 did not induce mTRPA1 activation but effectively activated hTRPA1. These results suggest that JT010 is a potent hTRPA1 agonist, but not mTRPA1.

Next, we examined the effects of JT010 on the TRPA1 channels expressed in human and mouse tissues. As inflammatory stimulation with interleukin-1α (IL-1α) transcriptionally induces TRPA1 expression in human FLSs [26,27], we applied JT010 and AITC to human FLSs with or without inflammation. As shown in Figure 3A, both 10 nM JT010 and 100 μM AITC induced substantial Ca^2+^ responses in inflammatory FLSs treated with IL-1α for 24 h, but not in control FLSs (vehicle). FLSs sensitive to AITC largely responded to JT010 (74%, 37 cells out of 50 cells) and vice versa (0%, 0 cells out of 40 cells), suggesting that JT010 can activate endogenous hTRPA1 channels in human tissues. In contrast, mouse DRGs sensitive to 100 μM AITC (40 cells out of 45 cells) did not respond to 10 nM JT010 (0 cells out of 40 cells), implying that JT010 at less than 10 nM could not activate endogenous mTRPA1 channels in mouse tissue (Figure 3B).

As previously reported [18,22], the substitution of cysteine in hTRPA1 at 621 (C621, Figure 4A) with serine (C621S) markedly reduced the response to 10 nM JT010, but not 100 μM AITC (Figure 4B–D, see also Figure 1B as the control), confirming that C621 is critical for JT010-induced TRPA1 response in humans. This C621 is conserved in mTRPA1 (C622, Figure 4A); therefore, we explored the insensitivity mechanism of mTRPA1 against JT010 using mutant mTRPA1s (Figure 4, Figure 5 and Figure 6). First, we focused on three cysteines in hTRPA1 that are not conserved in mTRPA1 (Y60, H1023, and N1027, arrow in Figure 4A). To test the possible involvement of cysteines in the sensitivity to JT010, we mutated these residues to cysteine (Y60C, H1023C, and N1027C) and applied JT010 to each mutant. None of these mutants were sensitive to JT010 at 10 nM, whereas all responded to 100 μM AITC (Figure 4B–D), indicating that these mTRPA1 residues may not determine the responsiveness to JT010.

A recent structural analysis of hTRPA1 revealed that the phenylalanine residue at position 669 (F669 shown in yellow in Figure 4A) of hTRPA1 is critical for channel activation by JT010 [22]. Indeed, this phenylalanine is not conserved in mTRPA1 (M670 in mouse; Figure 4A). In addition, the glutamate residue at position 668 (E668 shown in yellow in Fig4A) of hTRPA1 is substituted with serine in mTRPA1 (S669, Figure 4A). Therefore, we examined the involvement of these amino acid residues with different sensitivities to JT010 between hTRPA1 and mTRPA1. As shown in Figure 5A–C, the F669M mutation in hTRPA1 significantly reduced the sensitivity to JT010, but not to AITC, suggesting that M670 in mTRPA1 renders a lower sensitivity to JT010. Particularly, the mutant hTRPA1 with F669M was insensitive to 10 nM JT010 (Figure 5C). To further confirm the importance of F669, we applied JT010 to mTRPA1 with M670F mutation and double mutants with S669E and M670F mutations (Figure 6A,B). As 30 μM AITC induced a similar size of TRPA1 currents at +90 and −90 mV, the response of these mutants to JT010 did not differ from that of wild mTRPA1 (Figure 6C). Furthermore, the relative change in JT010-induced TRPA1 currents was analyzed to normalize the current amplitude against the control before application of JT010 for each TRPA1 (Figure 6D). The single M670F mutation in mTRPA1 (M670F-mTRPA1) was not sufficient to induce JT010-sensitivity. However, the double mutations of S669E and M670F induced weak but substantial sensitivity to 30 and 100 nM JT010 at +90 mV and 100 nM at −90 mV (Figure 6D). Nevertheless, the potency of this double mutant against JT010 was much lower than that of F669M- and wild-hTRPA1 (Figure 6D vs. Appendix A).

## 3. Discussion

In this study, we showed that the potent TRPA1 agonist JT010, which activated hTRPA1 at a concentration range of 0.3 to 100 nM, did not induce clear responses of mTRPA1 in HEK cells with heterologous expression of the channel. In contrast, both hTRPA1 and mTRPA1 showed similar responses to the conventional TRPA1 agonist, AITC. Moreover, JT010 induced the Ca^2+^ response of endogenous TRPA1 in human FLSs with inflammation but not in mouse DRG cells. As reported, substitution of F669 in the N-terminus of hTRPA1 to methionine, homologous to mTRPA1 methionine at 670, (F669M-hTRPA1) significantly reduced the response to JT010. In contrast, while a single M670F mutation in mTRPA1 was still insensitive to JT010, the double mutant of mTRPA1 with S669E and M670F mutations induced a weak but substantial response to JT010. Taken together, JT010 is a potent TRPA1 agonist in humans, but not in mice.

We confirmed that JT010, a potent TRPA1 agonist, is an effective hTRPA1 agonist. In experiments measuring Ca^2+^ responses and membrane ionic currents, 10–100 nM JT010 induced hTRPA1-dependent responses. Indeed, JT010 did not elicit a Ca^2+^ response in native HEK (Appendix A). Moreover, it has been reported that JT010, even at 1 μM, does not activate TRPV1, TRPV3, TRPV4, TRPM2, TRPM8, and TRPC5 [18]. While the EC_50_ of JT010 against hTRPA1 was 0.65 nM in a cell-based calcium uptake assay, it was ~7.6 nM in an electrophysiological study [18,22]. Comparing the pharmacological features of compounds using different assays can be challenging. In this study, the EC_50_ of JT010 was estimated to be 3–10 nM against hTRPA1 in Ca^2+^ measurements (Figure 1B) and electrophysiological assays (Figure 2 and Figure 5), strongly supporting that JT010 is a potent hTRPA1 agonist. Moreover, neither 10 nM JT010 nor 100 μM AITC elicited a Ca^2+^ response in human FLSs without inflammation. In contrast, both agonists evoked clear responses in the inflammatory FLSs, implying that endogenous TRPA1 in humans can be targeted by JT010. Consistently, injection of JT010 caused pain in humans with a half-maximal effective concentration of 0.31 μM [28], suggesting that JT010 is an effective TRPA1 agonist in vivo in human. In contrast, it has not been determined that JT010 is a weak TRPA1 agonist in vivo in rodents including mouse.

In this study, we confirmed that C621 and F669 are critical for the JT010-induced hTRPA1 activation. When we applied 10 nM JT010 to mutant hTRPA1 with C621S mutation, the Ca^2+^ response was abolished (Figure 4). Consistently, 10 nM JT010 failed to stimulate the Ca^2+^ response in C621S mutant cells [18]. Moreover, mutant hTRPA1 with C621S was insensitive to 100 nM JT010 in whole-cell current-recording experiments [22]. Therefore, it is reasonable to assume that the primary binding site of JT010 is the cysteine residue at position 621 of hTRPA1 (Figure 7). Meanwhile, based on the two-step model proposed, whereby C621 is the primary site of electrophile modification and C665 is another modification site for full channel activation, the binding of a small electrophile, AITC, to C665 in hTRPA1 could support the full activation of TRPA1. However, it has been proposed that bulky JT010 can stabilize the open pocket by modifying C621 alone [21]. It is pharmacologically useful to compare JT010 docking sites between hTRPA1 and mTRPA1. In our preliminary docking simulation, the affinity of JT010 against hTRPA1 was weak (ΔG = −5.9 kcal/mol). This low affinity cannot explain the high potency of JT010 against hTRPA1 in the previous and present experimental studies [18,22]. Because the docking sites at the highest rank simulated are different from those of the cryo-EM data, it is likely that the docking simulation is limited.

It is clear that mTRPA1 is less sensitive to JT010 than hTRPA1, and JT010 lower than 100 nM hardly activates mTRPA1. Intriguingly, mTRPA1 conserves both C622 (C621 in humans) and C666 (C665 in humans), which are critical cysteines for electrophilic modifications, including those of JT010 (Figure 4A). Suo et al. found that JT010 covalently binds to C621 of hTRPA1 and interacts with phenylalanine at 612 (F612) and tyrosine at 680 (Y680) of hTRPA1 via CH-π and sulfur-π formation through its thiazol group, respectively [22]. Moreover, the methoxyphenyl group of JT010 potentially interacts with histidine at 614 (H614), proline at 666 (P666), and F669. Mutants with serine at C621 (C621S), alanine at F612 (F612A), alanine at Y680 (Y680A), and alanine at F669 (F669A) mutations exhibited no sensitivity to 100 nM JT010. The importance of isoleucine at 623 (I623), tyrosine at 662 (Y662), and threonine at 684 (T684) of hTRPA1 against JT010 is also clear; the respective mutants dramatically reduce the response to JT010 [22]. Among these important residues, F669 alone is not conserved in mTRPA1, where the homologous residue is substituted with methionine (M670, Fig7), suggesting that this substitution lowers the sensitivity of mTRPA1 to JT010. While the mutant hTRPA1 with F669A mutation was insensitive to 100 nM JT010 [22], the mouse-type mutant with F669M in hTRPA1 retained the response to 30 and 100 nM JT010 (Figure 5). In contrast, the M670F mutation in mTRPA1 did not induce a clear JT010-dependent response (Figure 6). This suggests that F669 plays an important role in the response of hTRPA1 to JT010 and that M670 is not a critical determinant of lower sensitivity to JT010 in mTRPA1 (Figure 7).

Although the mTRPA1 double mutant with S669E and M670F mutations induced small responses to JT010, its potency was still significantly lower than that of hTRPA1 (Figure 6D vs. Appendix A). As the double mutant included all crucial amino acid residues for the interaction with JT010 proposed, it is notable that the interaction of JT010 with these residues is not sufficient to explain the activation mechanism of TRPA1 by lower JT010 concentration. In contrast, AITC (30 μM) induced large membrane currents of wild-type and all mutant TRPA1s in humans and mice in this study, indicating that large electrophiles like JT010 may have additional interactions with TRPA1. TRPA1 interacts differently with agonists and antagonists from species to species via distinctive molecular mechanisms. A96, a potent TRPA1 antagonist in humans and mice, is a TRPA1 agonist in chicken [29]. Moreover, menthol, a non-electrophile agonist of hTRPA1, inhibits mTRPA1 [23,30]. Particularly, when bulky electrophilic compounds are used, there may be species differences in the response of TRPA1. The basal channel activity was different between hTRPA1 and mTRPA1 under the intracellular dialysis of 0.3 μM Ca^2+^ (Figure 2B,D, 1520.2 ± 301.8 pA vs. 328.35 ± 74.73 pA in mTRPA1 and hTRPA1 at +90 mV, respectively, *p* < 0.01; −629.55 ± 210.1 pA vs. −87.38 ± 32.58 pA in mTRPA1 and hTRPA1 at −90 mV, respectively, *p* < 0.05). It is unlikely that this basal activity affected the interaction with JT010 in mTRPA1. Indeed, JT010 at concentrations under 100 nM did not affect mTRPA1, even in the absence of intracellular Ca^2+^, where the basal channel activity was lower (Appendix A). When applied to DRG cells isolated from mice, 10 nM of JT010 did not induce any response. Because 100 μM AITC evoked a response in 88% of cells employed, mTRPA1 expression was apparent, suggesting that a lower concentration of JT010 cannot activate endogenous mTRPA1. Notably, the application of 1000 nM JT010 induced a small but substantial Ca^2+^ response in HEK-mTRPA1 cells (Appendix A), possibly indicating the interaction of JT010 with C622 of mTRPA1. Nevertheless, it is clear that mTRPA1 is relatively resistant to lower JT010.

In conclusion, we showed that the potent hTRPA1 agonist JT010 could not activate mTRPA1 at concentrations ranging from 0.3 to 100 nM. Moreover, JT010 induced the response of endogenous TRPA1 in human FLSs with inflammation, but not in mouse DRG cells, both of which were sensitive to AITC. As confirmed by the importance of F669 in the N-terminus of hTRPA1 for the JT010-interaction, methionine substitution, which is homologous to mTRPA1 M670, significantly reduced the response to JT010. In contrast, while a single M670F mutation in mTRPA1 was still insensitive to JT010, the double mutant mTRPA1 with S669E and M670F mutations evoked a weak but substantial response to JT010. Taken together, JT010 is a potent TRPA1 agonist in humans, but not in mice.

## 4. Materials and Methods

### 4.1. Cell Culture

HEK cells obtained from the Health Science Research Resources Bank (HSRRB, Osaka, Japan) were maintained in Dulbecco’s modified Minimum Essential medium (D-MEM; Sigma-Aldrich, St. Louis, MO, USA) supplemented with 10% heat-inactivated fetal calf serum (FCS; Sigma-Aldrich), penicillin G (100 U/mL, Meiji Seika Pharma Co., Ltd., Tokyo, Japan), and streptomycin (100 μg/mL, Meiji Seika Pharma Co., Ltd., Tokyo, Japan). Human FLSs, which were purchased from Cell Applications (San Diego, CA, USA), were cultured in Synoviocyte Growth medium containing 10% growth supplement, 100 U/mL penicillin G, and 100 μg/mL streptomycin, as described previously [26]. FLSs were maintained at 37 °C in a 5% CO_2_ atmosphere. After they reached 70–80% confluence, FLSs were reseeded once every 10 days until nine passages were completed. The cells that grew with a doubling time of 6–8 days after this stage comprised a homogenous population, in which TRPA1 transcriptionally induced by IL-1αwas found to be unaffected. For the experiments, reseeded cells were cultured for 16 days and then exposed to IL-1α or vehicle.

### 4.2. Cell Isolation from DRG in Mice

This study was approved by the Animal Care Committee of Aichi Gakuin University (approval code 21-036 and 22-007) and conducted per the Guiding Principles for the Care and Use of Laboratory Animals approved by the Japanese Pharmacological Society. Male mice weighing 20–30 g were anesthetized with isoflurane and decapitated. Four to five DRGs isolated were washed in phosphate-buffered solution (PBS [in mM]: NaCl 137, KCl 5.4, MgCl_2_ 1.2, CaCl_2_ 2.2, Na_2_HPO_4_ 0.168, KH_2_PO_4_ 0.44, glucose 5.5, NaHCO_3_ 4.17, pH7.45) and treated with Ca^2+^-Mg^2+^-free PBS containing 0.05% collagenase (Amano, Nagoya, Japan) and 0.05% dispase (Boehringer Mannheim, Tokyo, Japan). All DRGs were kept in an incubator at 37 °C for 60 min and then the enzyme solution containing isolated DRGs was centrifuged at 1200× *g* rpm for 10 min. Thereafter, the supernatant was removed and the pellet was resuspended in culture medium (D-MEN with 10% FCS) and gently agitated with a fire-polished wide-pore pipette. Isolated DRG cells were allowed to attach to gelatin-coated glass coverslips in a 35 mm dish and were cultured for 24–48 h at 37 °C in a 5% CO_2_ atmosphere, and used within 48 h.

### 4.3. Recombinant Expression of Wild-Type and Mutant TRPA1 in HEK Cells

Partially confluent HEK cells (40–60% confluency) were transfected with the pcDNA3.1 and pcAc-GFP plasmids containing human and mouse TRPA1 using Lipofectamine 3000 (ThermoFisher Scientific, Yokohama, Japan). The TRPA1 mutants were constructed by PCR using mutant oligonucleotide primers in which the cysteine residue at amino acid positions 621 (C621) of hTRPA1 and tyrosine (Y60), histidine (H1023), and asparagine (N1027) residues at positions 60, 1023, and 1027 of mTRPA1 were replaced with serine (621S) and cysteine (60C, 1023C, and 1027C), respectively (Agilent Technologies, Santa Clara, CA, USA). The phenylalanine residue at position 669 (F669) of hTRPA1 and the methionine residue at position 670 (M670) of mTRPA1 were also mutated to methionine (F669M) and phenylalanine (M670F), respectively (Agilent Technologies). Additionally, a double mutant with mutation of serine at position 669 (S669) to glutamate (S669E) and M670F of mTRPA1 was also constructed. All experiments were performed within 48 h of transfection.

### 4.4. Western Blotting

HEK cells with TRPA1 were lysed in 50 μL lysis buffer ([in mM] Tris-HCl 50 (pH 8.0), NaCl 150, ethylenediaminetetraacetic acid (EDTA) 5) containing 1% NP-40, 0.5% sodium deoxycholate, 0.1% SDS, and protease inhibitors. The cell lysates were then incubated on ice for 30 min, vortexed every 5 min, and centrifuged at 20,000× *g* for 30 min at 4 °C. After each lysate (10 µg protein) was separated on an 8% polyacrylamide gel, proteins were transferred to a poly vinylidene difluoride (PVDF) membrane. Nonspecific binding of antibodies was blocked by incubation for 2 h in Tris-buffered saline (TBS) containing 5% skim milk and 0.1% Tween-20. The PVDF membrane was then incubated with the primary antibody (human-specific goat anti-TRPA1, production No.sc-32353, Santa Cruz Biotechnology Inc., Dallas, TX, USA, 1:1000 dilution) overnight at 4 °C. The blot was subsequently washed thrice with washing buffer (TBS containing 0.1% Tween-20), and a secondary antibody (IgG-HRP, 1:5000 dilution) was added to the PVDF membrane. Blots were washed again and detection reagents (Millipore) were added to generate a chemiluminescence signal. To determine the relative quantity of TRPA1 to β-actin in each sample, the PVDF membrane was exposed to a human anti-β-actin monoclonal antibody (production No. A5441; Sigma-Aldrich Inc., 1:5000 dilution). Finally, the membranes were scanned using a LAS-4000 apparatus (Fujifilm, Tokyo, Japan).

### 4.5. Patch-Clamp Experiments

Whole-cell current recordings were performed as previously described [31]. The resistance of the electrodes was 3–5 MΩ when filled with pipette solution. The Cs^+^-rich pipette solution contained [in mM] Cs-aspartate 110, CsCl 30, MgCl_2_ 1, 4-(2-hydroxyethyl)-1-piperazineethanesulfonic acid (HEPES) 10, ethylene glycol tetra-acetic acid (EGTA) 10, and Na_2_ATP 2 [adjusted to pH 7.2 with CsOH]. To maintain the activity of TRPA1 currents, intracellular free Ca^2+^ concentration was adjusted to a pCa value of 6.5 (0.3 μM Ca^2+^) by adding CaCl_2_ to the pipette solution. In some experiments, the EGTA concentration in the pipette solution was reduced to 1 mM in the absence of CaCl_2_. The membrane currents and voltage signals were digitized using an analog-digital converter (PCI-6229, National Instruments Japan, Tokyo, Japan). WinWCPV5.52 software was used for data acquisition and analysis of whole-cell currents (developed by Dr. John Dempster, University of Strathclyde, UK). The liquid junction potential between the pipette and bath solutions (−10 mV) was calculated. A ramp voltage protocol from −110 mV to +90 mV for 300 ms was applied every 5 s from a holding potential of −10 mV. The leak current component was not subtracted from the recorded current. A standard HEPES-buffered bathing solution (SBS [in mM]: NaCl 137, KCl 5.9, CaCl_2_ 2.2, MgCl_2_ 1.2, glucose 14, and HEPES 10 [adjusted to pH 7.4, with NaOH]) was used. All experiments were performed at 25 ± 1 °C.

### 4.6. Measurement of Ca^2+^ Fluorescence Ratio

HEK, FLSs, and DRG cells, which were loaded with 10 μM Fura2-AM (Dojindo, Kumamoto, Japan) in SBS for 30 min at room temperature, were superfused with SBS for 10 min, and Fura-2 fluorescence signals were measured at 0.1 Hz using the Argus/HisCa imaging system (Hamamatsu Photonics, Hamamatsu, Japan) driven by Imagework Bench 6.0 (INDEC Medical Systems, Santa Clara, CA, USA). Since the efficacy of gene transfection in HEK cells and the TRPA1 expression level in FLSs and DRG cells were similar but not identical from cell to cell, we collected 50, 5–11, and 8–12 single cells of HEK, FLSs, and DRG cells, respectively, on one coverslip to obtain the average response. We repeated the same protocol with other coverslips to obtain the mean and standard error of the mean (SEM) of independent experiments. In each analysis, the whole cell area was chosen as the region of interest to average the fluorescence ratio.

### 4.7. Modeling

Molecular modeling was performed on UCSF Chimera v1.16 [32] using the modeling software MODELLER v10.3 [33,34]. Using the amino acid sequence of mTRPA1 (ID: Q8BLA8) from UniProt (https://www.uniprot.org/ accessed on 30 August 2022), we modeled the structure of mTRPA1 as a monomer, with JT010-bound hTRPA1 (PBD ID:6PQO) as a template. Of the 100 models, the structure with the lowest normalized Discrete Optimized Protein Energy (zDOPE) score was adopted (Figure 7). The software Autodock-Vina was used to predict the possible binding models of JT010 to hTRPA1 (200 models) and the solutions were ranked according to their binding energy [35]. The grid box for docking model was set to locate C621 at the center and to include all amino acid residues interacted with JT010 (C665, F621, Y680, T684, Y662, I623, and F669 [22]).

### 4.8. Data Analysis

OriginJ9.1 was used for data analysis and representation. Data are expressed as individual values and mean ± SEM. Statistical significance between two groups and among multiple groups was examined using paired and unpaired Student’s *t*-tests (OrignJ9.1), two-way ANOVA (OrignJ9.1), Tukey–Kramer (OrignJ9.1), and Dunnett’s (KeyPlot5.0, KyensLab Inc., Tokyo, Japan) tests, respectively.

### 4.9. Reagents

The following drugs were used: acetylcholine chloride (ACh; Wako, Osaka, Japan), 2-chloro-N-(4-(4-methoxyphenyl)thiazol-2-yl)-N-(3-methoxypropyl-acetamide (JT010; Sigma-Aldrich), allyl isothiocyanate (AITC; Kanto Chemical Co., Tokyo, Japan), A-967079 (A96; FOCUS Biomolecules, Plymouth Meeting, PA, USA), and IL-1α (Sigma-Aldrich). Each drug was dissolved in a vehicle, as recommended by the manufacturer.

## Figures and Tables

**Figure 1 ijms-23-14297-f001:**
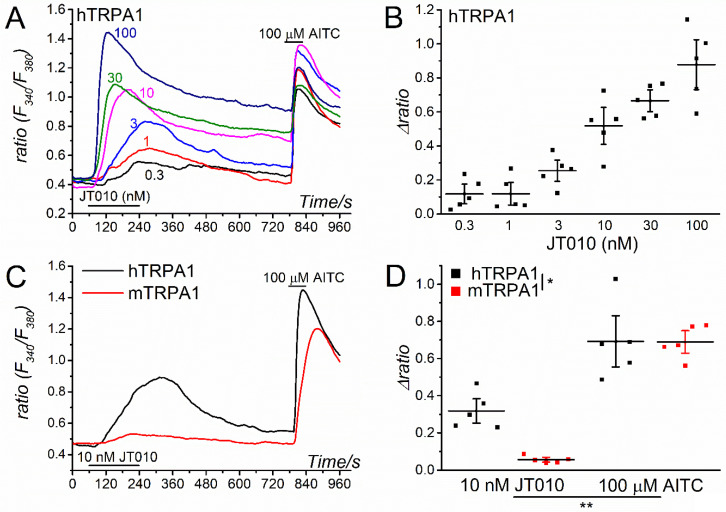
The hTRPA1-specific activation by JT010. (**A**,**B**) JT010 at a concentration range from 0.3 to 100 nM induced Ca^2+^ responses in representative HEK-hTRPA1 cells (**A**) and the peak JT010-induced Ca^2+^ response (Δratio) in HEK-hTRPA1 cells (five independent experiments) is summarized as a concentration-response relationship (**B**). At the end of each experiment, 100 μM AITC was applied to confirm hTRPA1 expression. (**C**,**D**) JT010 and AITC at 10 nM and 100 μM, respectively, were applied to HEK-hTRPA1 and HEK-mTRPA1 cells, and the measured Ca^2+^ response (**C**) and the peak JT010- and AITC-induced Ca^2+^ response (five independent experiments each) (**D**) are summarized. Two-way analysis of variance (ANOVA): * *p* = 0.0279, F = 5.85 (species); ** *p* < 0.0001, F = 84.5 (drugs); * *p* = 0.0312, F = 5.58 (interaction). Vertical bars = SEM.

**Figure 2 ijms-23-14297-f002:**
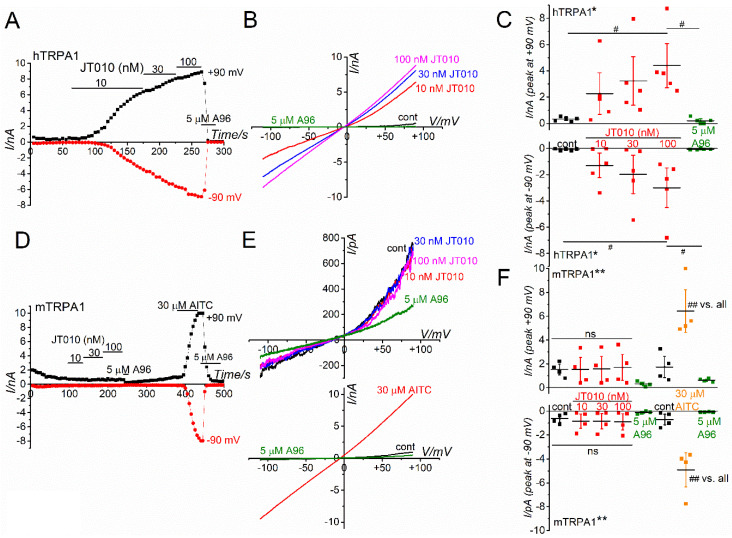
(**A**–**C**) JT010-induced membrane currents at −90 and +90 mV in a representative HEK-hTRPA1 cell which was superfused with SBS without Ca^2+^ (**A**) and the peak JT010-induced currents are summarized as a concentration-response relationship (**C**, five independent experiments). The current and voltage (I–V) relationships under each experimental condition are also shown (**B**). Repeated measures one-way ANOVA: * *p* = 0.0116. F = 4.28 (+90 mV). Post hoc Tukey–Kramer test: ^#^
*p* = 0.0285 (+90 mV vs. cont), ^#^
*p* = 0.0229 (+90 mV vs. A96). Repeated measures one-way ANOVA: * *p* = 0.0291. F = 3.37 (−90 mV). Post hoc Tukey–Kramer test: ^#^
*p* = 0.0495 (−90 mV vs. cont), ^#^
*p* = 0.0468 (−90 mV vs. A96). The pipette solution contained 0.3 μM Ca^2+^ to maintain TRPA1 activity. Ramp waveform voltage pulses from −110 to +90 mV for 300 ms were applied every 5 s at a holding potential of −10 mV. To confirm the hTRPA1 activity, cells were treated with A96 at 5 μM at the end of the agonist application. (**D**–**F**) No effects of JT010 on mTRPA1. The experimental conditions were identical to those demonstrated in (**A**), except HEK-mTRPA1 cell and additional applications of 30 μM AITC and 5 μM A96 (**D**). At the end of each experiment, AITC and A96 were applied to confirm the expression of mTRPA1. The peak JT010-and AITC-induced currents are summarized as a concentration-response relationship (**F**, four independent experiments). Repeated measures one-way ANOVA: ** *p* < 0.0001. F = 8.54 (+90 mV). Post hoc Tukey–Kramer test: ^##^
*p* < 0.0001 (+90 mV, vs. 1st A96 and 2nd A96), ^##^
*p* = 0.00038 (+90 mV, vs. cont), ^##^
*p* = 0.00039 (+90 mV, vs. 10 nM JT010), ^##^
*p* = 0.00045 (+90 mV, vs. 30 nM JT010), ^##^
*p* = 0.00062 (+90 mV vs. 100 nM JT010), ^##^
*p* = 0.00069 (+90 mV vs. 2nd cont). Repeated measures one-way ANOVA: ** *p* < 0.0001. F = 12.2 (−90 mV). Post hoc Tukey–Kramer test: ^##^
*p* < 0.0001 (−90 mV vs. all). The ‘ns’ shows no significance by the Tukey–Kramer test. Vertical bars = SEM.

**Figure 3 ijms-23-14297-f003:**
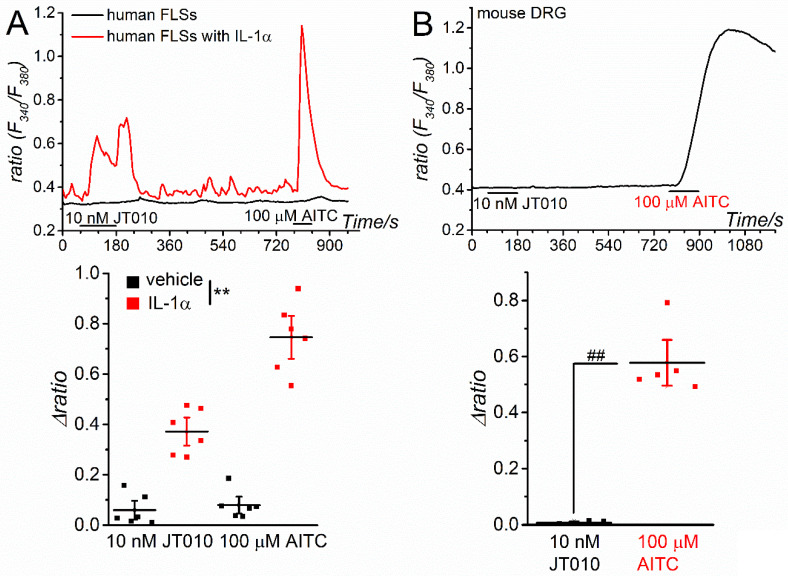
Effects of JT010 on endogenous TRPA1 in human and mouse cells. (**A**) JT010 at 10 nM and AITC at 100 μM were applied to human FLSs with or without inflammation. FLSs were treated with 10 U IL-1α or vehicle for 24 h and then exposed to JT010 and AITC, and Ca^2+^ response was monitored ((**A**), each representative cell). The peak JT010- and AITC-induced Ca^2+^ response (Δratio) in FLSs with or without IL-1α (six independent experiments each) are summarized in the lower panel. Two-way ANOVA: ** *p* < 0.0001, F = 167 (pretreatments); ** *p* < 0.0001, F = 27.0 (drugs); ** *p* = 0.00015, F = 21.7 (interaction) (**B**) JT010 at 10 nM and AITC at 100 μM were applied to mouse DRGs, and Ca^2+^ response was monitored ((**B**), a representative cell). The peak JT010- and AITC-induced Ca^2+^ response (Δratio) in DRGs are summarized (lower panel, five independent experiments). Paired Student’s *t*-test: ^##^
*p* = 0.00042. Vertical bars = SEM.

**Figure 4 ijms-23-14297-f004:**
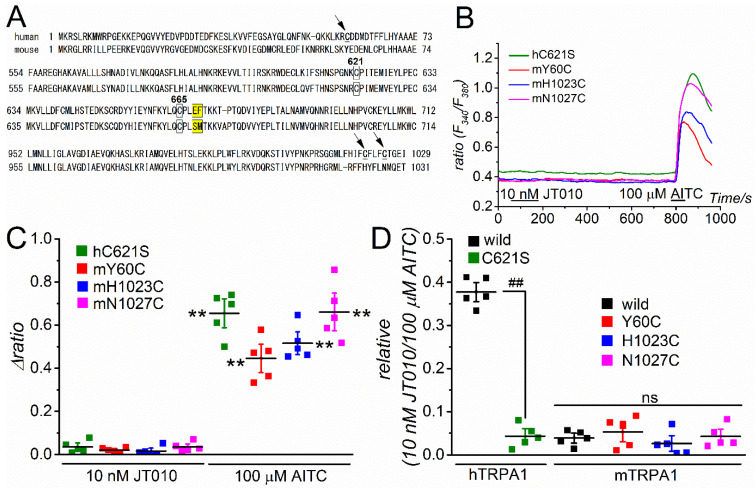
Comparison of JT010-induced TRPA1 response among mutants of N- and C-terminal cysteine residues of hTRPA1 and mTRPA1. (**A**) Alignment of amino acid sequence between hTRPA1 and mTRPA1. C621 and C665 in hTRPA1 (homologous to mTRPA1 C622 and C666) shown by boxes indicate critical cysteines for electrophilic TRPA1 agonist modification. Bold and underlined letters (C59, C1021, C1025 in human, indicated by an arrow) show cysteines substituted in mutant mTRPA1 (Y60C, H1023C, N1027C), whose effect was examined. Yellow color boxes indicate potential critical amino acids for JT010-sensitivity, whose importance is examined in Figure 5 and Figure 6. (**B**–**D**) Ca^2+^ responses of mutant hTRPA1 with C621S mutation and mutant mTRPA1s with Y60C, H1023C, and N1027C mutations to 10 nM JT010. To confirm the channel expression, 100 µM AITC was applied at the end of the experiment. Each representative Ca^2+^ response was superimposed (**B**) and the peak JT010- and AITC-induced Ca^2+^ response (Δratio) is summarized (**C**, five independent experiments). Paired Student’s *t*-test: ** *p* < 0.0001, ** *p* = 0.00092, ** *p* < 0.0001, and ** *p* = 0.00064 for hC621S, mY60C, mH1023C, and mN1027C, respectively (**D**) Ca^2+^ response to JT010 was normalized with that to AITC and is summarized. The responses of wild hTRPA1 and mTRPA1 were also included as a comparison (the same data set as Figure 1). Unpaired Student’s *t*-test: ^##^
*p* < 0.0001. The ‘ns’ shows no significance by the Tukey–Kramer test. Vertical bars = SEM.

**Figure 5 ijms-23-14297-f005:**
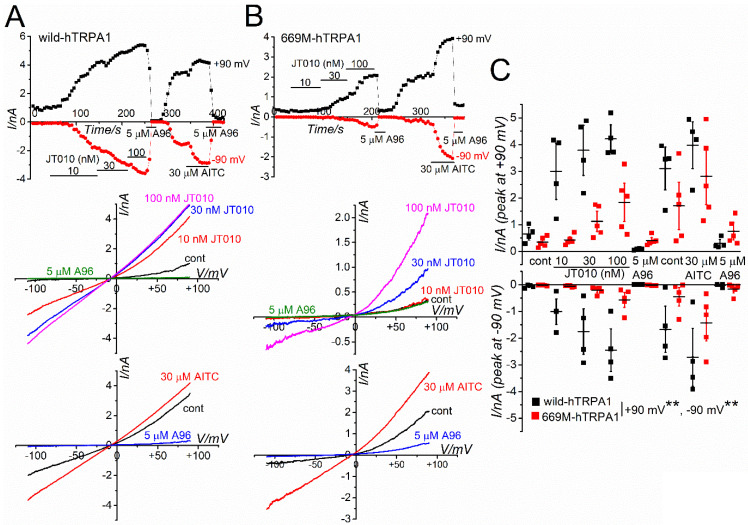
Critical importance of F669 for JT010-induced hTRPA1 response. (**A**–**C**) JT010 (10 to 100 nM)- and AITC (30 µM)-induced currents at −90 and +90 mV were compared between hTRPA1 (**A**) and 669M-hTRPA1 (**B**), and are summarized (**C**, four-five independent experiments). Following agonist treatment, 5 µM A96 was added to block TRPA1 channel current components. In the lower panel of (**A**,**B**), I–V relationships under each experimental condition are shown. Cells were superfused with SBS without Ca^2+^ and dialyzed with Cs-aspartate rich pipette solution including 0.3 µM Ca^2+^. Ramp waveform voltage pulses from −110 to +90 mV for 300 ms were applied every 5 s. Two-way ANOVA (+90 mV): ** *p* < 0.0001, F = 33.3 (hTRPA); ** *p* < 0.0001, F = 17.1 (drugs); ** *p* = 0.00045, F = 4.54 (interaction). Two-way ANOVA (−90 mV): ** *p* < 0.0001, F = 28.6 (hTRPA); ** *p* < 0.0001, F = 11.4 (drugs); ** *p* = 0.00846, F = 3.06 (interaction). Vertical bars = SEM.

**Figure 6 ijms-23-14297-f006:**
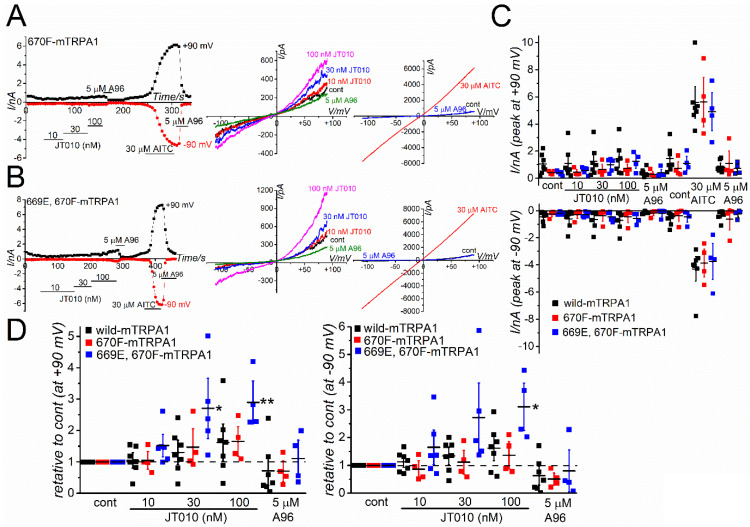
S669 and M670 are the potential amino acids that determine the low sensitivity of mTRPA1 to JT010. Cells were superfused with SBS without Ca^2+^ and dialyzed with a Cs-aspartate-rich pipette solution including 0.3 µM Ca^2+^. Ramp waveform voltage pulses from −110 to +90 mV for 300 ms were applied every 5 s. (**A**,**B**) JT010 was commutatively applied to HEK cells with an M670F substitution in mTRPA1 (**A**, 670F-mTRPA1) and double substitutions of S669E and M670F (**B**, 669E, 670F-mTRPA1) to examine the effects on membrane currents at −90 and +90 mV, and the pooled data of the peak currents evoked are summarized (**C**, four to six independent experiments including the same data set as in Figure 2F). After applying 100 nM JT010, 5 µM A96 was added to block the TRPA1 channel current components. For comparison, 30 µM AITC was used. In the middle and right panels of (**A**,**B**), the I-V relationships under each experimental condition are shown. (**D**) Each current amplitude shown in (**C**) was normalized to that of the control without JT010 and exhibited the relative amplitude change under each treatment. Dunnett’s multiple comparisons test was performed for each TRPA1 gene. * *p* = 0.0116 and ** *p* = 0.00829 for 30 and 100 nM JT010, respectively in 669E, 670F-mTRPA1 (+90 mV). * *p* = 0.0327 for 100 nM JT010 in 669E, 670F-mTRPA1 (−90 mV). Vertical bars = SEM.

**Figure 7 ijms-23-14297-f007:**
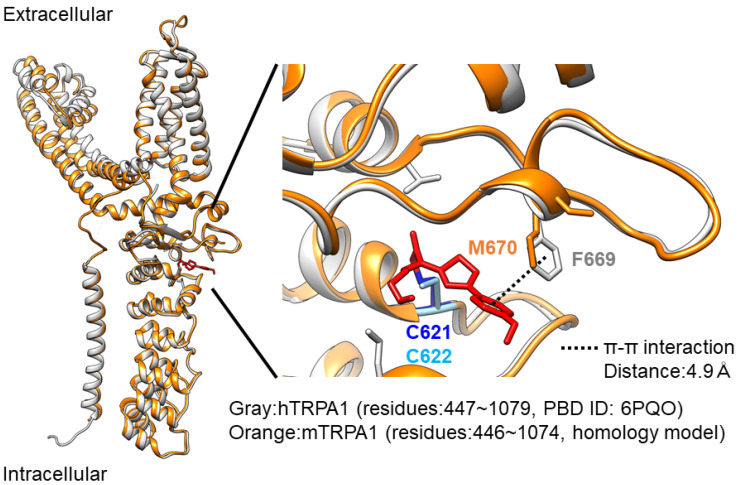
Structure modeling of the interaction of JT010 with F669 on hTRPA1 (PDB ID: 6PQO) and M670 on mTRPA1. Ribbon diagrams colored grey of the hTRPA1 atomic model for residues 447–1079 are shown on the left. The Modeller optimized model of mTRPA1 for residues 446–1074 and JT010 are colored orange and red, respectively. A close-up view of the JT010 binding sites is shown on the right. The methoxyphenyl group of JT010 potentially interacts with the phenyl of F669 in hTRPA1 in a π-π interaction manner. The coordination is shown by the dotted line with a 4.9 Å distance. Out of 100 models, the structure with the lowest zDOPE score (2.69) was adopted (see also Materials and Methods).

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
