# Peer review of "Potent Activation of Human but Not Mouse TRPA1 by JT010"

_ijms, 2022, doi:10.3390/ijms232214297_

Round 1

Reviewer 1 Report

MS # ijms-2007212

Very specific TRPA1 agonists are lacking. The present paper is interesting with that subject. In addition, this molecule may be human specific contrary to PF-4840154 for example. However, out of the six papers studying JT010 in the literature, only one examined the in-vivo effect of JT010 in human (Heber et al. J Neuroscience 2019; 39:3845) and none in rodents. This should be discussed briefly.

Abbreviations. Many abbreviations are not defined, either in the text or in the specific section.

P2L64. Please replace potent by highly selective.

P2L93; isoflurane, not isoflavones

Molecular modeling. This part may be expanded. Docking with the calculation of the Gibbs free energy of binding might have been calculated, thus leading to comparisons with EC50/IC50/Kd already published (Takaya et al., ref 18, Samanta et al. J Gen Physiol 2018 May 7;150(5):751 and other).

Figure 1D. I presume that *interaction refers to ANOVA. This is unclear. In addition, the visual difference between dots and stars is unclear (same for all figures).

Author Response

Dear Editor,

Re: Revision of ijms-2007212

Enclosed please find a revised manuscript titled " Potent activation of human but not mouse TRPA1 by JT010" by Matsubara et al., to IJMS.

Here, we provide our responses to the peer review comments, revising the manuscript where necessary. We have responded, point-by-point, to the reviewers' comments. Our responses are in bold text in this letter and the changes in revised manuscript are shown in revised MS with Tack Changes. In particular, because the resolution of figures was not sufficient as pointed by both reviewers, we have improved them except Fig.7. We hope the changes are satisfactory.

Sincerely,

Professor Katsuhiko Muraki, Ph.D.

*Address Correspondence to: Prof Katsuhiko Muraki, Laboratory of Cellular Pharmacology, School of Pharmacy, Aichi-Gakuin University, 1-100 Kusumoto, Chikusa, Nagoya 464-8650, Japan. Fax:+81-52-757-6799; E-mail: [email protected]

Responses to the peer review comments

Dear Reviewer1: Thank you very much for your remarks and for your time. We have made concerted efforts to address your comments and revise the manuscript accordingly. We have responded, point-by-point, to your comments. Our responses are in bold text in this letter and the changes in revised manuscript are shown in text in MS with Track Changes.

  1. However, out of the six papers studying JT010 in the literature, only one examined the in-vivo effect of JT010 in human (Heber et al. J Neuroscience 2019; 39:3845) and none in rodents. This should be discussed briefly.

Thank you for your valuable comment. We discussed this point in Discussion in revised MS (lines 396-399, P12, please see below).

Consistently, injection of JT010 caused pain in humans with a half-maximal effective concentration of 0.31 μM, suggesting that JT010 is an effective TRPA1 agonist in vivo in human. In contrast, it has not been determined that JT010 is a weak TRPA1 agonist in vivo in rodents including mouse.

  1. Abbreviations. Many abbreviations are not defined, either in the text or in the specific section.

Thank you for your comment. We carefully checked all abbreviations in the text in revised MS.

  1. P2L64. Please replace potent by highly selective..

Thank you for your suggestion. We replaced potent by highly selective in revised MS (line 65, P2).

  1. P2L93; isoflurane, not isoflavones

Thank you for your comment. We amended it in revised MS (line 98, P3).

  1. Molecular modeling. This part may be expanded. Docking with the calculation of the Gibbs free energy of binding might have been calculated, thus leading to comparisons with EC50/IC50/Kd already published (Takaya et al., ref 18, Samanta et al. J Gen Physiol 2018 May 7;150(5):751 and other).

Thank you very much for your valuable comment. Preliminary, we have performed the docking simulation of JT010 against hTRPA1 to use Autodock-vina. Unfortunately, the Gibbs free energy of binding between JT010 and hTRPA1 is not sufficient to explain the high potency. However, because this is informative for readers, we added some discussion (lines 410-416, P12) and hence we amended Method and Materials (lines 176-181, P4) to explain the Docking simulation.

Methods and Materials:

The software Autodock-Vina was used to predict the possible binding models of JT010 to hTRPA1 (200 models) and the solutions were ranked according to their binding energy. The grid box for docking model was set to locate C621 at the center and to include all amino acid residues interacted with JT010 (C665, F621, Y680, T684, Y662, I623, and F669).

Discussion:

It is pharmacologically useful to compare JT010 docking sites between hTRPA1 and mTRPA1. In our preliminary docking simulation, the affinity of JT010 against hTRPA1 was weak (ΔG=-5.9 kcal/mol). This low affinity cannot explain the high potency of JT010 against hTRPA1 in the previous and present experimental studies. Because the docking sites at the highest rank simulated are different from those of the cryo-EM data, it is likely that the docking simulation is limited.

  1. Figure 1D. I presume that *interaction refers to ANOVA. This is unclear. In addition, the visual difference between dots and stars is unclear (same for all figures).

Thank you very much for your useful comments. We totally agree with you. We improved figure legends to explain ANOVA interaction in revised MS (Fig.1 and Fig3). In addition, all figures were enlarged and circle symbols were replaced with squares in revised MS.

In addition, we added a Graphical Abstract to revised MS and improved the animal welfare procedure.

Reviewer 2 Report

The manuscript “Potent activation of human but not mouse TRPA1 by JT010” by Matsubara et al is a research article which tested the effects of the synthetic compound JT010 on human TRPA1 (hTRPA1) and mouse TRPA1 (mTRPA1) expressed in human kidney (HEK) cells. The authors clearly show that JT010 potentially and selectively activates hTRPA1 but not mTRPA1. Generally, the subject is of interest and scientifically sound and contains essential contents. This paper is also of importance for providing us the novel and important pharmacological evidence of the JT010. The manuscript has been well organized and written. However, I have minor concerns on the paper.

The statistical significance was assessed using Student t-tests and two-way ANOVA. The values of “t” and “F” should be given if possible.

Figure 1, 2, 4, 5, 6: The letter size is too small to understand the figures.

Author Response

Dear Editor,

Re: Revision of ijms-2007212

Enclosed please find a revised manuscript titled " Potent activation of human but not mouse TRPA1 by JT010" by Matsubara et al., to IJMS.

Here, we provide our responses to the peer review comments, revising the manuscript where necessary. We have responded, point-by-point, to the reviewers' comments. Our responses are in bold text in this letter and the changes in revised manuscript are shown in revised MS with Tack Changes. In particular, because the resolution of figures was not sufficient as pointed by both reviewers, we have improved them except Fig.7. We hope the changes are satisfactory.

Sincerely,

Professor Katsuhiko Muraki, Ph.D.

*Address Correspondence to: Prof Katsuhiko Muraki, Laboratory of Cellular Pharmacology, School of Pharmacy, Aichi-Gakuin University, 1-100 Kusumoto, Chikusa, Nagoya 464-8650, Japan. Fax:+81-52-757-6799; E-mail: [email protected]

Responses to the peer review comments

Responses to the peer review comments

Dear Reviewer2: Thank you very much for your remarks and for your time. We have made concerted efforts to address your comments and revise the manuscript accordingly. We have responded, point-by-point, to your comments. Our responses are in bold text in this letter and the changes in revised manuscript are shown in text in MS with Track Changes.

  1. The statistical significance was assessed using Student t-tests and two-way ANOVA. The values of “t” and “F” should be given if possible.

Thank you very much for your valuable comment. We added t and F values in each figure legend in revised MS.

  1. Figure 1, 2, 4, 5, 6: The letter size is too small to understand the figures.

Thank you very much for your useful comment. We totally agree with you. All figures were enlarged and circle symbols were replaced with squares in revised MS.

In addition, we added a Graphical Abstract to revised MS and improved the animal welfare procedure.